# Omni-SILA: Towards Omni-scene Driven Visual Sentiment Identifying, Locating and Attributing in Videos

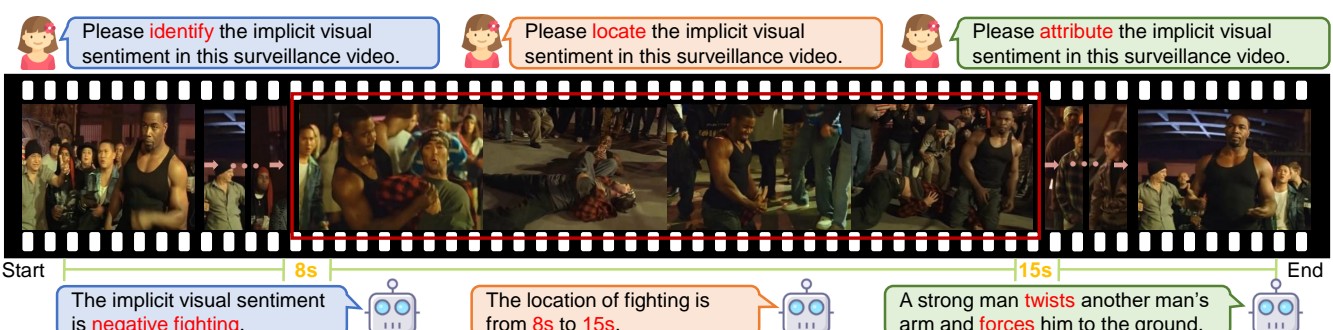

**Figure 1: A sample from our constructed implicit Omni-SILA dataset to illustrate the Omni-SILA task, where the proposed ICM approach is required to identify, locate and attribute the negative implicit visual sentiment *fighting* in this surveillance video.**

## Abstract

Prior studies on Visual Sentiment Understanding (VSU) primarily rely on the explicit scene information (e.g., facial expression) to judge visual sentiments, which largely ignore implicit scene information (e.g., human action, objection relation and visual background), while such information is critical for precisely discovering visual sentiments. Motivated by this, this paper proposes a new **Omni**-scene driven visual **S**entiment **I**dentifying, **L**ocating and **A**ttributing in videos (Omni-SILA) task, aiming to interactively and precisely identify, locate and attribute visual sentiments through both explicit and implicit scene information. Furthermore, this paper believes that this Omni-SILA task faces two key challenges: modeling scene and highlighting implicit scene beyond explicit. To this end, this paper proposes an **I**mplicit-enhanced **C**ausal **M**oE (ICM) approach for addressing the Omni-SILA task. Specifically, a **S**cene-**B**alanced **M**oE (SBM) and an **I**mplicit-**E**nhanced **C**ausal (IEC) blocks are tailored to model scene information and highlight the implicit scene information beyond explicit, respectively. Extensive experimental results on our constructed explicit and implicit Omni-SILA datasets demonstrate the great advantage of the proposed ICM approach over advanced Video-LLMs.

## CCS Concepts

• **Computing methodologies** → **Artificial intelligence**.

## Keywords

Omni-Scene Information, Implicit-enhanced Causal MoE Framework, Visual Sentiment Identifying, Locating and Attributing

## 1 Introduction

Visual Sentiment Understanding (VSU) [54, 68] focuses on leveraging explicit scene information (e.g., facial expression) to understand the sentiments of images or videos. However, there exist many surveillance videos in the real world, where implicit scene information (e.g., human action, object relation and visual background)

can more truly reflect visual sentiments compared with explicit scene information. In light of this, this paper defines the need to rely on implicit scene information to precisely identify visual sentiments as implicit visual sentiments, such as robbery, shooting and other negative implicit visual sentiments under surveillance videos. More importantly, current VSU studies mainly focus on identifying the visual sentiments, yet they ignore exploring when and why these sentiments occur. Nevertheless, this information is critical for sentiment applications, such as effectively filtering negative or abnormal contents in the video to safeguard the mental health of children and adolescents [10, 38, 45].

Building on these considerations, this paper proposes a new **Omni**-scene driven visual **S**entiment **I**dentifying, **L**ocating and **A**ttributing in videos (Omni-SILA) task[1], which leverages Video-centred Large Language Models (Video-LLMs) for interactive visual sentiment identification, location and attribution. This task aims to identify what is the visual sentiment, locate when it occurs and attribute why this sentiment through both explicit and implicit scene information. Specifically, the Omni-SILA task identifies, locates and attributes the visual sentiment segments through interactions with LLM. As shown in Figure 1, a strong man is fighting another man during the timestamps from 8s to 15s, where the LLM is asked to identify, locate and attribute this *fighting* implicit visual sentiment. In this paper, we explore two major challenges when leveraging Video-LLMs to comprehend omni-scene (i.e., both explicit and implicit scene) information for addressing the Omni-SILA task.

On one hand, how to model explicit and implicit scene information is challenging. Existing Video-LLMs primarily devote to modeling general visual information for various video understanding tasks. Factually, while LLMs encode vast amounts of world knowledge, they lack the capacity to perceive scenes [27, 33]. Compared to general visual information, explicit and implicit scene

---

[1]**Relevance to the Web**: Omni-SILA task belongs to *Enhancement of multimedia* topic of *Semantics and Knowledge* track, which aims to analyze sentiment from web content, such as videos from YouTube and Bilibili. The works involved in this paper aim to make web content more harmless and helpful for online browsers and communication.

information is crucial in the Omni-SILA task. Taking Figure 1 as an example, the negative implicit visual sentiment *fighting* in the video is clearly conveyed through the action *twists an arm* and *forces to ground*. However, due to the heterogeneity of these omni-scene information (i.e., various model structures and encoders), a single, fixed-capacity transformer-based model fails to capitalize on this inherent redundancy, making Video-LLMs difficult to capture important scene information. Recently, MoE has shown scalability in multi-modal heterogeneous representation fusion tasks [41]. Inspired by this, we take advantage of the MoE architecture to model explicit and implicit scene information in videos, thereby evoking the omni-scene perceptive ability of Video-LLMs.

On the other hand, how to highlight the implicit scene information beyond the explicit is challenging. Since explicit scene information (e.g., facial expression) has richer sentiment semantics than implicit scene information (e.g., subtle actions), it is easier for models to model explicit scene information, resulting in modeling bias for explicit and implicit scene information. However, compared with explicit scene information, implicit scene information has more reliable sentiment discriminability and often reflects real visual sentiments as reported by Lian et al. [28]. For the example in Figure 1, the strong man is *laughing* while *twists another man's arm* and *forces him to the ground*, where the facial expression *laughing* contradicts the negative *fighting* visual sentiment conveyed by the actions of *twists the arm* and *forces to ground*. Recently, causal intervention [35] has shown capability in mitigating biases among different information [57]. Inspired by this, we take advantage of causal intervention to highlight the implicit scene information beyond the explicit, thereby mitigating the modeling bias to improve a comprehensive understanding of visual sentiments.

To tackle the above challenges, this paper proposes an **I**mplicit-enhanced **C**ausal **M**oE (ICM) approach, aiming to identify, locate and attribute visual sentiments in videos. Specifically, a **S**cene-**B**alanced **M**oE (SBM) module is designed to model both explicit and implicit scene information. Furthermore, an **I**mplicit-**E**nhanced **C**ausal (IEC) module is tailored to highlight the implicit scene information beyond the explicit. Moreover, this paper constructs two explicit and implicit Omni-SILA datasets to evaluate the effectiveness of our ICM approach. Comprehensive experiments demonstrate that ICM outperforms several advanced Video-LLMs across multiple evaluation metrics. This justifies the importance of omni-scene information for identifying, locating and attributing visual sentiment, and the effectiveness of ICM for capturing such information.

## 2  Related Work

• **Visual Sentiment Understanding.** Previous studies on Visual Sentiment Understanding (VSU) utilize multiple affective information to predict the overall sentiment of images [54, 70] or videos [60, 63]. For image, traditional studies focus on extracting sentiment features to analyze sentiments [53, 55, 69], while recent studies focus on using instructions to fine-tune LLMs to precisely predict sentiments [51]. For videos, traditional studies require pre-processing video features and predicting video sentiments by elaborating complex fusion strategies [14, 43, 44, 62] or learning superior representations [16, 17, 56, 61]. To achieve end-to-end goal, some studies [2, 49, 68] input the entire videos, and explore the location of

segments that convey different sentiments or anomalies. Recently, a few studies gradually explore the causes of anomalies [9] and sentiments [28] via Video-LLMs. However, these efforts have not addressed visual sentiment identification, location and attribution of videos at the same time. Different from all the above studies, this paper proposes a new Omni-SILA task to interactively answer what, when and why are the visual sentiment through omni-scene information, aiming to precisely identify and locate, as well as reasonably attribute visual sentiments in videos.

• **Video-centred Large Language Models.** Recently, large language models (LLMs) [34], such as LLaMA [42] and Vicuna [5], have shown remarkable abilities in NLP area. Given the multimodal nature of the world, some studies [3, 23, 30, 73] have explored using LLMs to enhance visual understanding. Building on these, Video-LLMs have extended into the more sophisticated video area, enabling them to process complex interactions between videos and instructions. According to the role of LLMs, Video-LLMs be broadly categorized into three types. **(1)** LLMs as text decoders means LLMs receive embeddings from the video encoder and decode them into text outputs based on instructions, including Video-ChatGPT [32], Video-LLaMA [65], Valley [31], Otter [22], mPLUG-Owl [59], Video-LLaVA [29], Chat-UniVi [18], VideoChat [24] and MiniGPT4-Video [1]. **(2)** LLMs as regressors means LLMs can predict continuous values, like bounding boxes for object trajectories, including TimeChat [37], GroundingGPT [26], HawkEye [47] and Holmes-VAD [66]. **(3)** LLMs as hidden layers means LLMs do not directly output text but connect to a designed task-specific head to perform tasks, like event time localization, including OneLLM [13], VITRON [11] and GPT4Video [48]. Although the aforementioned Video-LLMs studies make significant progress in video understanding, they remain limitations in their ability to perceive omni-scene information and are unable to analyze harmful video content. Therefore, this paper proposes the ICM approach, aiming to evoke the omni-scene perception capabilities of Video-LLMs and highlight the implicit scenes beyond the explicit.

## 3  Approach

In this section, we formulate our Omni-SILA task as follows. Given a video $v$ consisting of $N$ segments, each segment $n$ is labeled with a time $t$, visual sentiment $s$ and cause $c$. The goal of Omni-SILA is to interactively identify what is the visual sentiment, locate when it occurs, and attribute why it arises within $v$. Thus, the model generates a set of segments $\{(t_1, s_1, c_1), ..., (t_i, s_i, c_i), ..., (t_n, s_n, c_n)\}$, where $t_i$, $s_i$ and $c_i$ denote the time, visual sentiment and cause for each video segment.

In this paper, we propose an **I**mplicit-enhanced **C**ausal **M**oE (ICM) approach to address the Omni-SILA task, which involves two challenges: modeling scene and highlighting implicit scene beyond explicit. To address these challenges, we design a **S**cene-**B**alanced **M**oE (SBM) block and an **I**mplicit-**E**nhanced **C**ausal (IEC) block. Particularly, we choose the open-sourced Video-LLaVA [29] as the backbone, which achieves state-of-the-art performance on most video understanding benchmarks. The overall framework is shown in Figure 2. Prior to delving into the intricacies of the core components within ICM, we provide an overview of the encoding block of each scene information.

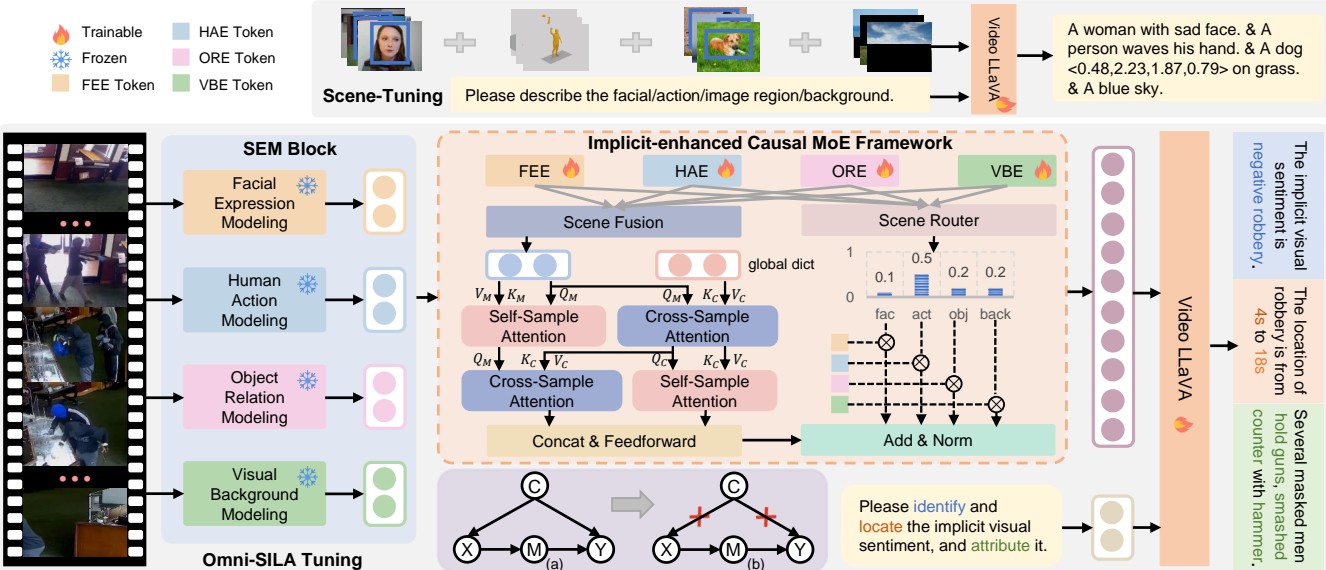

**Figure 2: The overall architecture of our ICM approach, consisting of a Scene-Enriched Modeling (SEM) block and an Implicit-enhanced Causal MoE framework, which comprises a Scene-Balanced MoE (SBM) block (right, see Section 3.2) and an Implicit-Enhanced Causal (IEC) block (left, see Section 3.3), where (a) and (b) are causal graphs for IEC block. FEE, HAE, ORE and VBE represent Facial Expression Expert, Human Action Expert, Object Relation Expert and Visual Background Expert.**

## 3.1 Scene-Enriched Modeling Block

Given a set of video segments $v = [v_1, ..., v_i, ..., v_n]$, we leverage four blocks to capture explicit and implicit scene information as shown in Figure 2. **Facial Expression Modeling** is used to model and capture explicit facial expression from individuals by MTCNN [67], which is a widely-used network to detect and learn the representation of facial expression. Specifically, MTCNN first detects face candidate proposals in each video segments $v_i$, and then produces the final face detection results and key point locations, which are encoded by CNN to obtain the facial expression representation $x_f$. **Human Action Modeling** is used to model and capture implicit human action from individuals by HigherHRNet [4], which is a well-studied network to learn scale-aware action representation. Specifically, HigherHRNet detects the location of action key points or parts (e.g., elbow, wrist, etc) for each individual in each video segment $v_i$, and employs HRNet [46] and designs a deconvolutional module to generate higher-resolution action heatmaps to obtain the human action representation $x_a$. **Object Relation Modeling** is used to model and capture implicit object relations from each video segment $v_i$ by RelTR [6], which is a well-studied model to learn the object relations representation. Specifically, RelTR generates the relations between subjects and objects, such as *<man, holds, gun>*, and extracts the visual feature context and entity representations to obtain object relations representation $x_o$. **Visual Background Modeling** is used to model and capture implicit visual backgrounds from each video segment $v_i$ by ViT [7] and SAM-V1 [20], which are two advanced visual encoding and segmenting tools. Specifically, we leverage SAM-V1 to segment the visual backgrounds of each segment $v_i$, with pure black to fill out the masked parts. Then we transform these processed segments into ViT to obtain the final visual background representation $x_b$.

## 3.2 Scene-Balanced MoE Block

In this study, we take advantage of MoE [41] architecture and design a **S**cene-**B**alanced **M**oE (SBM) block to model scene information. Specifically, we address two crucial questions: (1) how to model different types of scene information; (2) how to balance the contributions of different scene information for the Omni-SILA task. We will provide comprehensive answers to these two questions in the subsequent section, formulated as follows.

**Scene Experts** are introduced to answer question (1), which model both explicit and implicit scene information inspired by Han et al. [13], consisting of Facial Expression Expert (FEE), Human Action Expert (HAE), Object Relation Expert (ORE) and Visual Background Expert (VBE) four scene experts. Each scene expert is a stack of transformer layers, aiming to dynamically learn different scene information. As shown in Figure 2, unlike other MoE approaches that embed several FFNs within LLMs, our four scene experts operate externally to the LLM, enabling effective alignment of various scene information. Formally, for the representations $x_i$, $i \in \{f, a, o, b\}$ of the four scene modeling blocks, the output representation $h_i$ of each expert $\text{Expert}_i$ can be denotes as: $h_i = \text{Expert}_i(x_i)$, where $\text{Expert}_i$ represents the general term of FEE (f), HAE (a), ORE (o) and VBE (b) four scene experts.

**Balanced MoE** is leveraged to answer question (2), which balances different scene information contributions, managed by a dynamic scene router $R$ as shown in Figure 2. Balanced MoE is structured as a straightforward MLP that processes input features $h$ of four scene experts and computes routing weights for each expert, effectively functioning as a soft router [36]. Formally, the output $y_{\text{moe}}$ of the balanced MoE can be denoted as follows:

$$y_{\text{moe}} = \text{LayerNorm}(\sum_{j=1}^{L} g_j(h) \times E_j(h)) \qquad (1)$$

where $g_j(h)$ and $E_j(h)$ denote the corresponding weight and the output of the $j$-th scene expert, and $L$ is the number of scene experts.

To obtain $g(h)$, we computer the gating probability $P$ of each scene expert for input $h$, formulated as follows:

$$g(h) = \mathbf{P} = \text{softmax}(\mathbf{W} \cdot \mathbf{h}) \tag{2}$$

where $\mathbf{W} \in \mathbb{R}^{L \times d}$ is a learnable parameter for scene router $R$, and $d$ is the hidden dimension of each expert. $\mathbf{P}$ is a vector size $L$ and $\mathbf{P_j}$ denotes the probability of the $j$-th scene expert $E_j$ to process $h$.

Furthermore, to optimize the scene router $R$, we design a router loss with balancing constraints $\mathcal{L}_{\text{rb}}$, encouraging $R$ to dynamically adjust the contributions of all scene experts, formulated as:

$$\mathcal{L}_{\text{rb}} = -\alpha \cdot \sum_{j=1}^{L} \mathbf{P}_j * \log(\mathbf{P}_j) + \beta \cdot L \cdot \sum_{j=1}^{L} \mathbf{G}_j * \mathbf{H}_j \tag{3}$$

The first term with the hyper-parameter $\alpha$ measures the contribution of various scene information, encouraging the scene router $R$ to assign a different weight to each scene expert within the constraints of $\mathbf{P}$, thereby preventing $R$ from uniformly assigning the same weight and leading to wrong visual sentiment judgment. We expect the routing mechanism to select the scene experts that are more important for the Omni-SILA task, thus we minimize the entropy of the gating probability distribution $\mathbf{P}$ to ensure that each input feature $h_i$ could be assigned the appropriate weight coefficient. The second term with the hyper-parameter $\beta$ balances scene experts of different sizes (since the output dimension $d$ of four scene modeling blocks are different), forcing the model not to pay too much attention to scene experts with high dimensions, while ignoring scene experts with low dimensions during the learning process. $\mathbf{G}_j = \frac{1}{L} \sum_{j=1}^{L} \mathbb{1}\{e_j \in E_j\} \times d$ represents the average dimension of the hidden state of the scene expert $e_j$ on the entire input $h$, which imports the influence of scene expert sizes $d$ when the model focuses more on large scene experts, the loss rises, which direct the model to more economically utilize smaller scene experts. $\mathbf{H}_j = \frac{1}{L} \sum_{j=1}^{L} \mathbf{P}_j$ represents the gating probability assigned to $e_j$.

## 3.3 Implicit-Enhanced Causal Block

In this study, we take advantage of the causal intervention technique [35] and design an **I**mplicit-**E**nhanced **C**ausal (IEC) block to highlight implicit scene beyond explicit. Specifically, there are also two crucial questions to be answered: (1) how to highlight implicit scene information through the front-door adjustment strategy [35]; (2) how to implement this front-door adjustment strategy in the Omni-SILA task. Next, we will answer the two questions.

**Causal Intervention Graph** is introduced to answer question (1), which formulates the causal relations among the scene information X, the fusion scene features M, visual sentiment outputs Y, and confounding factors C as shown in Figure 2 (a). In this graph, X → M → Y represents the desired causal effect from the scene information X to visual sentiment outputs Y, with the fusion scene features M serving as a mediator. X ← C → Y represents the causal effect of the invisible confounding factors C on both scene information X and visual sentiment outputs Y.

To highlight implicit scene information, we consider mitigating the modeling bias between X and C present in the path M → Y, thus we leverage $do$-operator [35] to block the back-door path M ← X ← C → Y through conditioning on X as shown in Figure 2

(b). Then, we utilize the front-door adjustment strategy to analyze the causal effect of X → Y, denoted as: $P(\text{Y} = y|do(\text{X} = x)) = \sum_m P(m|x) \sum_x P(x)[P(y|x,m)]$.

**Deconfounded Causal Attention** is leveraged to answer question (2), which implements the front-door adjustment strategy through the utilization of attention mechanisms. Given the expensive computational cost of network forward propagation across all samples, we use the Normalized Weighted Geometric Mean (NWGM) [39, 52] approximation. Therefore, we sample X, M and compute $P(\text{Y}|do(\text{X}))$ through feeding them into the network, and then leverage NWGM approximation to achieve the goal of deconfounding explicit and implicit scene biases, represented as follows:

$$P(\text{Y}|do(\text{X})) \approx \text{softmax}[f(y_x, y_m)] \tag{4}$$

where $f(.)$ followed by a softmax layer is a network, which is used to parameterize the predictive distribution $P(y|x, m)$. In addition, $y_m = \sum_m P(\text{M} = m|p(\text{X}))\boldsymbol{m}$ and $y_x = \sum_x P(\text{X} = x|q(\text{X}))\boldsymbol{x}$ estimate the self-sampling and cross-sampling respectively, where the variables $m, x$ correspond to the embedding vectors of $\boldsymbol{m}, \boldsymbol{x}$. $p(.)$ and $q(.)$ are query embedding functions parameterized as networks, which are used to transform the input X into two distinct query sets. Therefore, we utilize the attention mechanism to estimate the self-sampling $y_m$ and cross-sampling $y_x$ as shown in Figure 2:

$$y_m = \begin{cases} \mathbf{V}_M \cdot \text{softmax}(\mathbf{Q}_M^\top \mathbf{K}_M) \\ \mathbf{V}_C \cdot \text{softmax}(\mathbf{Q}_C^\top \mathbf{K}_C) \end{cases} \tag{5}$$

where Eq.(5) denotes self-sampling attention to compute intrinsic effect of fusion scene features M and confounding factors C.

$$y_x = \mathbf{V}_C \cdot \text{softmax}(\mathbf{Q}_M^\top \mathbf{K}_C) \tag{6}$$

where Eq.(6) represents the cross-sampling attention to compute the mutual effect between the fusion scene features M and confounding factors C. In the implementation of two equations, $\mathbf{Q}_M$ and $\mathbf{Q}_C$ are derived from $p(\text{X})$ and $q(\text{X})$. $\mathbf{K}_M$ and $\mathbf{V}_M$ are obtained from the current input sample, while $\mathbf{K}_C$ and $\mathbf{V}_C$ come from other samples in the training set, serving as the global dictionary compressed from the whole training dataset. Specifically, we initialize this dictionary by using K-means clustering [15] on all the embeddings of samples in the training set. To obtain the final output $y_{iec}$ of the IEC block, we employ an FFN to integrate the self-sampling estimation $y_m$ ans cross-sampling estimation $y_x$, formulated as: $y_{iec} = \text{FFN}(y_m + y_x)$.

## 3.4 Two-Stage Training Optimization

Due to the lack of scene perception abilities in Video-LLaVA, we design a two-stage training process, where scene-tuning stage is pre-tuned to perceive omni-scene information, while Omni-SILA tuning stage is trained to address the Omni-SILA task better via the perception abilities of scene information, detailed as follows.

For **Scene-Tuning** stage, we utilize four manually annotated instruction datasets (detailed in Section 4.1) to pre-tune Video-LLaVA, aiming to evoke the scene perception abilities of Video-LLaVA, where the model is asked to "*Please describe the facial/action/image region/background*". For **Omni-SILA Tuning** stage, we meticulously construct an Omni-SILA dataset (detailed in Section 4.1) to make our ICM approach better tackling the Omni-SILA task through instruction tuning, where the ICM approach is asked through the instruction "*Please identify and locate the implicit visual sentiment,*

**Table 1: Comparison of several Video-LLMs and our ICM approach on Explicit and Implicit Omni-SILA dataset for identifying and locating sentiments. The ↓ beside FNRs indicates the lower the metric, the better the performance. Bold and underlined indicate the highest and second-highest performance, respectively (the same below).**

| Approach | Explicit Omni-SILA Dataset | | | | | | | Implicit Omni-SILA Dataset | | | | | | |
| | Acc | F2 | FNRs↓ | mAP@IoU | | | | Acc | F2 | FNRs↓ | mAP@IoU | | | |
| | | | | 0.1 | 0.2 | 0.3 | Avg | | | | 0.1 | 0.2 | 0.3 | Avg |
| mPLUG-Owl | 60.33 | 59.57 | 71.37 | 30.30 | 12.20 | 3.36 | 15.29 | 28.88 | 30.06 | 73.98 | 31.42 | 13.21 | 4.46 | 16.36 |
| PandaGPT | 64.22 | 64.12 | 49.12 | 28.28 | 17.17 | 7.98 | 17.81 | 32.48 | 33.87 | 49.62 | 29.36 | 18.28 | 8.87 | 18.83 |
| Valley | 65.75 | 65.01 | 56.07 | 31.35 | 15.15 | 6.76 | 17.75 | 34.66 | 35.94 | 53.49 | 32.24 | 16.26 | 7.66 | 18.75 |
| VideoChat | 66.57 | 65.80 | 44.50 | 30.93 | 20.62 | 8.25 | 22.63 | 35.12 | 36.44 | 50.79 | 31.96 | 21.73 | 9.26 | 20.98 |
| Video-ChatGPT | 67.88 | 66.84 | 61.26 | 25.56 | 18.89 | 10.00 | 18.15 | 37.82 | 39.31 | 61.47 | 26.65 | 19.91 | 11.03 | 19.19 |
| ChatUniVi | 67.23 | 66.57 | 61.81 | 18.82 | 10.61 | 9.05 | 12.83 | 37.95 | 38.88 | 62.52 | 19.89 | 11.62 | 10.02 | 13.84 |
| Video-LLaVA | 68.19 | 67.08 | 44.32 | 31.41 | 15.78 | 8.82 | 18.67 | 40.02 | 41.88 | 50.34 | 32.41 | 16.79 | 9.92 | 19.71 |
| **ICM** | **71.41** | **70.21** | **33.38** | **31.91** | **23.39** | **18.75** | **25.21** | **47.39** | **48.36** | **32.76** | **34.79** | **26.14** | **19.08** | **27.88** |
| w/o SBM | 69.32 | 68.36 | 37.92 | 30.33 | 22.23 | 15.59 | 22.72 | 43.18 | 44.52 | 40.11 | 32.44 | 23.49 | 15.65 | 23.68 |
| w/o IEC | 69.71 | 68.82 | 35.85 | 31.27 | 23.23 | 16.53 | 23.68 | 44.12 | 45.08 | 38.62 | 33.18 | 24.20 | 16.23 | 24.54 |
| w/o scene-tuning | 67.87 | 66.32 | 43.59 | 26.80 | 18.51 | 12.05 | 19.12 | 39.64 | 40.76 | 49.74 | 27.88 | 19.65 | 13.14 | 20.23 |

*and attribute it*" as shown in Figure 2. Note that the instruction will be text tokenized inside Video-LLaVA to obtain the textual token $y_t$, which is added with the normalized combination of the SBM block output $y_{moe}$ and IEC block output $y_{iec}$. Thus, the input of the LLM inside Video-LLaVA will be "Norm$(y_{moe} + y_{iec}) + y_t + y_v$", where $y_v$ denotes the visual features encoded by intrinsic visual encoder LanguageBind [72] of Video-LLaVA. Moreover, the whole loss of our ICM approach can be represented as $\mathcal{L} = \mathcal{L}_{lm} + \mathcal{L}_{rb}$, where $\mathcal{L}_{lm}$ is the original language modeling loss of the LLM.

## 4 Experimental Settings

### 4.1 Datasets Construction

To assess the effectiveness of our ICM approach for the Omni-SILA task, we construct instruction datasets for two stages.

For **Scene-Tuning** stage, we aim to improve Video-LLaVA's ability to understand scenes, including facial expression, human action, object relations and visual backgrounds. Specifically, we choose four datasets, CMU-MOSEI [63], HumanML3D [12], Ref-COCO [19] and Place365 [71], to manually construct instructions for each video or image. For instance in Figure 2, with the instruction "*Please describe the facial/action/image region/background*", the responses are "*A woman with sad face./A person waves his hand./A dog <0.48,2.23,1.87,0.79> on the grass./A blue sky.*". Particularly, we use SAM-V2 [20] to segment objects and capture visual backgrounds in Place365. Since CMU-MOSEI and HumanML3D contain over 20K videos, we sample frames at an appropriate rate to obtain 200K frames. To ensure scene data balance, we randomly select 200K images from RefCOCO and Place365.

For **Omni-SILA Tuning** stage, we aim to train our ICM approach to address the Omni-SILA task through instruction tuning. We construct an Omni-SILA tuning dataset consisting of 202K video clips, and we sample 8 frames for each video clip, resulting in 1.62M frames. This dataset consists of an explicit Omni-SILA dataset (training: 52K videos, test: 25K videos) and an implicit Omni-SILA dataset[2] (training: 102K videos, test: 23K videos). The explicit

---

[2]Implicit Omni-SILA dataset contains mainly audio-free surveillance video, and the open-sourced Video-LLMs generally do not support audio. Therefore, to ensure model consistency and fair comparison, this paper focuses on visual sentiment understanding.

Omni-SILA dataset is based on public TSL-300 [68], which contains explicit *positive*, *negative* and *neutral* three visual sentiment types. Due to its lack of sentiment attributions, we leverage GPT-4V [58] to generate the description of each frame from four aspects: facial expression, human action, object relations and visual backgrounds, and then again use GPT-4V to summarize visual sentiment attributions. Finally, we manually check and adjust inappropriate attributions. Implicit Omni-SILA dataset is based on public CUVA [9], which contains implicit *Fighting* (1), *Animals Hurting People* (2), *Water Incidents* (3), *Vandalism* (4), *Traffic Accidents* (5), *Robbery* (6), *Theft* (7), *Traffic Violations* (8), *Fire* (9), *Pedestrian Incidents* (10), *Forbidden to Burn* (11), *Normal* twelve visual sentiment types. Further, we manually construct instructions for each video clip. Specifically, with the beginning of instruction "*You will be presented with a video. After watching the video*", we ask the model to identify "*please identify the explicit/implicit visual sentiments in the video*", locate "*please locate the timestamp when ...*", and attribute "*please attribute ... considering facial expression, human action, object relations and visual backgrounds*" visual sentiments. The corresponding responses are "*The explicit/implicit visual sentiment is ...*", "*The location of ... is from 4s to 18s.*", "*The attribution is several ...*".

For inference, we evaluate the effectiveness of our ICM approach on both explicit and implicit Omni-SILA datasets. Particularly, due to the goal of identifying, locating and attributing visual sentiments, we infer three tasks with the same instruction in Omni-SILA tuning stage as described before.

### 4.2 Baselines

Due to the requirement of interaction and pre-processing videos, traditional VSU approaches are not directly suitable for our Omni-SILA task. Therefore, we choose several advanced Video-LLMs as baselines. **mPLUG-Owl** [59] equips LLMs with multimodal abilities via modular learning. **PandaGPT** [40] shows impressive and emergent cross-modal capabilities across six modalities: image/video, text, audio, depth, thermal and inertial measurement units. **Valley** [31] introduces a simple projection to unify video, image and language modalities with LLMs. **VideoChat** [24] designs a VideoChat-Text module to convert video streams into text and a VideoChat-Embed module to encode videos into embeddings. **Video-ChatGPT** [32]

**Table 2: Comparison of several Video-LLMs and our ICM approach on Explicit and Implicit Omni-SILA datasets for visual sentiments attributing, where GPT-based and Human indicate two methods to evaluate the Atr-R metric.**

| Approach | Explicit Omni-SILA Dataset | | | | | Implicit Omni-SILA Dataset | | | | |
|---|---|---|---|---|---|---|---|---|---|---|
| | Sem-R | Sem-C | Sen-A | Atr-R | | Sem-R | Sem-C | Sen-A | Atr-R | |
| | | | | GPT-based | Human | | | | GPT-based | Human |
| mPLUG-Owl | 0.216 | 42.06 | 53.23 | 6.57 | 2.92 | 0.506 | 59.65 | 69.68 | 5.51 | 2.09 |
| PandaGPT | 0.231 | 43.12 | 56.72 | 6.73 | 3.08 | 0.516 | 60.47 | 72.65 | 5.68 | 2.68 |
| Valley | 0.252 | 45.41 | 57.93 | 6.94 | 3.36 | 0.535 | 63.36 | 70.72 | 6.07 | 2.46 |
| VideoChat | 0.243 | 45.06 | 58.16 | 7.06 | 3.51 | 0.527 | 63.79 | 71.65 | 6.29 | 2.95 |
| Video-ChatGPT | 0.272 | 48.55 | 60.54 | 7.39 | 3.89 | 0.558 | 65.53 | 75.37 | 6.75 | 3.42 |
| ChatUniVi | 0.254 | 46.69 | 59.33 | 7.24 | 3.72 | 0.532 | 63.57 | 74.54 | 6.87 | 3.15 |
| Video-LLaVA | 0.266 | 47.27 | 61.40 | 7.95 | 4.05 | 0.547 | 64.45 | 74.12 | 7.04 | 3.38 |
| **ICM** | **0.290** | **54.79** | **65.38** | **9.02** | **4.95** | **0.599** | **73.57** | **81.94** | **8.89** | **4.74** |
| w/o SBM | 0.275 | 49.76 | 63.44 | 8.36 | 4.21 | 0.561 | 68.07 | 77.22 | 7.51 | 3.77 |
| w/o IEC | 0.280 | 50.22 | 63.62 | 8.52 | 4.26 | 0.567 | 69.65 | 78.66 | 7.87 | 3.95 |
| w/o scene tuning | 0.252 | 45.92 | 60.53 | 7.76 | 3.96 | 0.539 | 63.79 | 73.63 | 6.86 | 3.19 |

combines the capabilities of LLMs with a pre-trained visual encoder optimized for spatio-temporal video representations. **ChatUniVi** [18] uses dynamic visual tokens to uniformly represent images and videos, and leverages multi-scale representations to capture both high-level semantic concepts and low-level visual details. **Video-LLaVA** [29] aligns the representation of images and videos to a unified visual feature space, and uses a shared projection layer to map these unified visual representations to the LLMs.

### 4.3 Implementation Details

Since the above models target different tasks and employ different experimental settings, for a fair and thorough comparison, we re-implement these models and leverage their released codes to obtain experimental results on our Omni-SILA datasets. In our experiments, all the Video-LLMs size is 7B. The hyper-parameters of these baselines remain the same setting reported by their public papers. The others are tuned according to the best performance. For ICM approach, during the training period, we use AdamW as the optimizer, with an initial learning rate 2e-5 and a warmup ratio 0.03. We fine-tune Video-LLaVA (7B) using LoRA for both scene-tuning stage and Omni-SILA stage, and we set the dimension, scaling factor, dropout rate of the LoRA matrix to be 16, 64 and 0.05, while keeping other parameters at their default values. The parameters of MTCNN, HigherHRNet, RelTR and ViT are frozen during training stages. The number of experts in Causal MoE is 4, and the layers of each expert are set to be 8. The hyper-parameters $\alpha$ and $\beta$ of $\mathcal{L}_{rb}$ are set to be 1e-4 and 1e-2. ICM approach is trained for three epochs with a batch size of 8. All training runs on 1 NVIDIA A100 GPU with 40GB GPU memory. It takes around 18h for scene-tuning stage, 62h for training Omni-SILA stage and 16h for inference. To facilitate the corresponding research in this direction, all codes together with datasets will be released.

### 4.4 Evaluation Metrics

To comprehensively evaluate the performance of various models on the Omni-SILA task, we use commonly used metrics and design additional task-specific ones. We categorized these evaluation metrics into three tasks, as described below.

• **Visual Sentiment Identifying (VSI)**. We leverage Accuracy (Acc) to evaluate the performance of VSI following Yang et al. [56]. Besides, we prioritize Recall over Precision and report F2-score [68].

• **Visual Sentiment Locating (VSL)**. Following prior studies [21, 64], we use mAP@IoU metric to evaluate VSL performance. This metric is calculated as the mean Average Precision (mAP) under different intersections over union (IoU) thresholds (0.1, 0.2 and 0.3). More importantly, we emphasize false-negative rates (FNRs) [25], denoted as: $\frac{number\ of\ false-negative\ frames}{number\ of\ positive\ frames}$, which refer to the rates of "*misclassifying a positive/normal frame as negative*". FNRs indicate that it is preferable to classify all timestamps as negative than to miss any timestamp associated with negative sentiments, as this could lead to serious criminal events.

• **Visual Sentiment Attributing (VSA)**. We design four specific metrics to comprehensively evaluate the accuracy and rationality of generated sentiment attributions. Specifically, semantic relevance (Sem-R) leverages the Rouge score to measure the relevance between generated attribution and true cause. Semantic consistency (Sem-C) leverages cosine similarity to assess the consistency of generated attribution and true cause. Sentiment accuracy (Sen-A) calculates the accuracy between generated attribution and true sentiment label. Attribution rationality (Atr-R) employs both automatic and human evaluations to assess the rationality of generated attributions. For automatic evaluation, we use ChatGPT [34] to score based on two criteria: sentiment overlap and sentiment clue overlap, on a scale of 1 to 10. For human evaluation, three annotators are recruited to rate the rationality of the generated attributions on a scale from 1 to 6, where 1 denotes "*completely wrong*" and 6 denotes "*completely correct*". After obtaining individual scores, we average all the scores to report as the final results. Moreover, $t$-test[3] is used to evaluate the significance of the performance.

## 5 Results and Discussion

### 5.1 Experimental Results

Table 1 and Table 2 show the performance comparison of different approaches on our Omni-SILA task (including VSI, VSL and VSA). From this table, we can see that: **(1)** For VSI, our ICM approach outperforms all the Video-LLMs on the implicit Omni-SILA dataset, and achieves comparable results on the explicit Omni-SILA dataset. For instance, compared to the best-performing Video-LLaVA, ICM achieves average improvements by 6.93% ($p$-value<0.01) on implicit

---

[3]https://docs.scipy.org/doc/scipy/reference/stats.html

**Table 3: The effectiveness study of various scenes in Omni-SILA, where ✓ means that we capture the current scene. All the experiments are conducted on Explicit and Implicit Omni-SILA datasets and evaluate VSI, VSL and VSA three tasks. Fac, Act, Obj and Back are short for facial expression, human action, object relation and visual background scene, respectively.**

| Fac | Act | Obj | Back | Explicit Omni-SILA Dataset | | | | | | Implicit Omni-SILA Dataset | | | | | |
|-----|-----|-----|------|-------|-------|-------|-----------|-------|-------|-------|-------|-------|-----------|-------|-------|
| | | | | Acc | F2 | FNRs↓ | mAP@tIoU | Sen-A | Atr-R | Acc | F2 | FNRs↓ | mAP@tIoU | Sen-A | Atr-R |
| | ✓ | ✓ | ✓ | 68.81 | 67.79 | 36.65 | 23.25 | 63.29 | 12.38 | 46.18 | 45.93 | 34.52 | 25.77 | 79.26 | 12.70 |
| ✓ | | ✓ | ✓ | 68.66 | 67.87 | 37.52 | 22.64 | 63.48 | 12.16 | 43.89 | 44.97 | 37.89 | 24.62 | 77.78 | 11.36 |
| ✓ | ✓ | | ✓ | 70.03 | 69.06 | 34.97 | 24.07 | 64.15 | 13.01 | 45.34 | 46.22 | 34.54 | 26.31 | 79.41 | 12.39 |
| ✓ | ✓ | ✓ | | 69.16 | 68.32 | 36.07 | 23.01 | 63.62 | 12.66 | 44.57 | 45.49 | 35.86 | 25.39 | 78.36 | 11.78 |
| ✓ | ✓ | ✓ | ✓ | 71.41 | 70.21 | 33.38 | 25.51 | 65.38 | 13.97 | 47.39 | 48.36 | 32.76 | 27.88 | 81.94 | 13.63 |

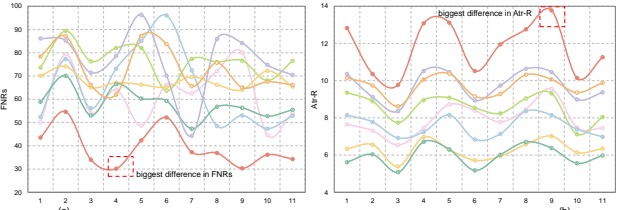

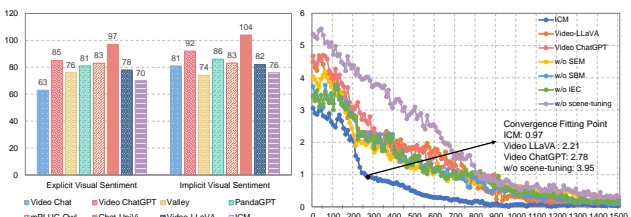

**Figure 3: Two line charts to compare several well-performing Video-LLMs with our ICM approach on 11 implicit visual sentiments of FNRs (a) and Atr-R (b) two metrics, and the red boxes indicate the categories *Vandalism* of FNRs and *Fire* of Atr-R where the performance difference is biggest.**

**Figure 4: Two statistical charts to illustrate the efficiency of our ICM approach. The histogram (a) compares the inference time of ICM with baselines, while the line chart (b) shows the convergence of training losses of ICM, two well-performing Video-LLMs and the variants of ICM across training steps.**

Omni-SILA dataset and 3.18% ($p$-value<0.05) on explicit Omni-SILA dataset. This indicates that identifying implicit visual sentiments is more challenging than the explicit, and justifies the effectiveness of ICM in identifying what is visual sentiment. **(2)** For VSL, similar to the results on VSI task, our ICM approach outperforms all the baselines on the implicit Omni-SILA dataset while achieves comparable results on the explicit. For instance, compared to the best-performing results underlined, ICM achieves the average improvements by 5.44% ($p$-value<0.01) on the implicit and 3.65% ($p$-value<0.05) on the explicit. Particularly, our ICM approach surpasses all the Video-LLMs on FNRs by 17.58% ($p$-value<0.01) on the implicit and 10.94% ($p$-value<0.01) on explicit compared with the best results underlined. This again justifies the challenge in locating the implicit visual sentiments, and demonstrates the effectiveness of ICM in locating when the visual sentiment occurs. **(3)** For VSA, our ICM approach outperforms all the Video-LLMs on both implicit and explicit Omni-SILA datasets. Specifically, compared to the best-performing approach on all VSA metrics, ICM achieves total improvements of 5.9%, 14.28%, 10.55% and 5.18 on Sem-R, Sem-C, Sen-A and Atr-R in two datasets. Statistical significance tests show that these improvements are significant ($p$-value<0.01). This demonstrates that ICM can better attribute why are both explicit and implicit visual sentiments compared to advanced Video-LLMs, and further justifies the importance of omni-scene information.

## 5.2 Contributions of Key Components

To further study the contribution of the key components in our ICM approach, we conduct a series of ablations studies, the results of which are detailed in Table 1 and Table 2. From these tables, we can see that: **(1) w/o IEC** block shows inferior performance compared to ICM, with an average decrease of VSI, VSL and VSA three tasks by 4.82% ($p$-value<0.05), 6.6% ($p$-value<0.01) and 10.37% ($p$-value<0.01).

This indicates the existence of bias between explicit and implicit information, and further justifies the effectiveness of IEC block to highlight the implicit scene information beyond the explicit. **(2) w/o SBM** block shows inferior performance compared to ICM, with an average decrease of VSI, VSL and VSA three tasks by 5.99% ($p$-value<0.01), 9.29% ($p$-value<0.01) and 13.12% ($p$-value<0.01). This indicates the effectiveness of SBM block in modeling and balancing the explicit and implicit scene information via the MoE architecture, encouraging us to model heterogeneous information via MoE. **(3) w/o scene tuning** exhibits obvious inferior performance compared to ICM, with an average decreases of VSI, VSL and VSA three tasks by 11.39% ($p$-value<0.01), 20.47% ($p$-value<0.01) and 23.72% ($p$-value<0.01). This confirms that the backbone lacks the ability to understand omni-scene information. This further demonstrates the necessity and effectiveness of pre-tuning, and encourages us to introduce more high-quality datasets to improve the scene understanding ability of Video-LLMs.

## 5.3 Effectiveness Study of Scene Information

To delve deeper into the impact of various scene information, we conduct a series of ablations studies, the results of which are detailed in Table 3. From this table, we can see that: **(1) w/o Facial Expression Modeling** shows more obvious inferior performance on the explicit than the implicit Omni-SILA dataset, with the total decrease of VSI, VSL and VSA by 5.02% ($p$-value<0.01), 5.53% ($p$-value<0.01) and 3.68% ($p$-value<0.05) on the explicit; 3.64% ($p$-value<0.05), 3.87% ($p$-value<0.05) and 3.61% ($p$-value<0.05) on the implicit. This is reasonable that most of videos in the implicit dataset may have no visible faces, making it difficult to capture facial expression. **(2) w/o Human Action Modeling** exhibits obvious inferior performance on both explicit and implicit Omni-SILA datasets, with the total decrease of VSI, VSL and VSA by 5.99% ($p$-value<0.01), 7.7% ($p$-value<0.01) and 5.07% ($p$-value<0.01). This indicates that human

action information is important to recognize explicit and implicit visual sentiments. For example, we can precisely identify, locate and attribute the *theft* via *an obvious human action of stealing*. **(3) w/o Object Relation Modeling** shows slight inferior performance on both explicit and implicit Omni-SILA datasets. This is reasonable that objects are objective exists, and the visual sentiment of visual object relations is very subtle, unless a specific scene like *<a man, holding, guns>* can be identified as *robbery*. **(4) w/o Visual Background Modeling** exhibits obvious inferior performance on both explicit and implicit Omni-SILA datasets, with the total decrease of VSI, VSL and VSA by 4.92%($p$-value<0.05), 5.39%($p$-value<0.01) and 4.25%($p$-value<0.05). This indicates that video background is also important to recognize explicit and implicit visual sentiments. For example, we can precisely identify, locate and attribute the *fire* via the *forest on fire with flames raging to the sky* background.

## 5.4 Applicative Study of ICM Approach

To study the applicability of ICM, we compare the FNRs and Atr-R of ICM with other Video-LLMs. From Table 1, we can see that ICM approach performs the best on the metric of FNRs. For example, ICM outperforms the best-performing Video-LLaVA by 10.94% ($p$-value<0.01) and 17.58% ($p$-value<0.01) on the explicit and implicit Omni-SILA dataset respectively. From Table 2, our ICM approach achieves state-of-the-art performance on both implicit and explicit Omni-SILA datasets. These results indicate that ICM is effective in reducing the rates of FNRs and providing reasonable attributions, which is of great importance in application. Furthermore, recognizing that the implicit Omni-SILA dataset comprises 11 distinct real-world crimes, we perform a detailed analysis of each negative implicit visual sentiment on the performance of FNRs (Figure 3 (a)) and Atr-R (Figure 3 (b))[4]. From these figures, we can see that ICM surpasses all other Video-LLMs across 11 crimes. Particularly, ICM performs best on *Vandalism* (4) in FNRs and *Fire* (9) in Atr-R. This indicates that ICM is effective in reducing FNRs and improving the interpretability of negative implicit visual sentiments, encouraging us to consider the omni-scene information, such as *human action* in *Vandalism* and *visual background* in *Fire*, for precisely identifying, locating and attributing visual sentiments.

## 5.5 Efficiency Analysis of ICM Approach

To study the efficacy of our ICM approach, we compare the inference time of ICM with other Video-LLMs (Figure 4 (a)), and analyze the convergence of training loss for two advanced Video-LLMs (i.e., Video-ChatGPT and Video-LLaVA), ICM and its variants over different training steps (Figure 4 (b)). As shown in Figure 4 (a), we can see that ICM achieves little difference in inference time compared with other Video-LLMs. This is reasonable because MoE can improve the efficiency of inference [8, 50], which encourages us to model omni-scene information via MoE architecture. From Figure 4 (b), we can see that: **(1)** ICM shows fast convergence compared to Video-LLMs. At the convergence fitting point, the loss of ICM is 0.97, while Video-LLaVA is 2.21. This indicates that ICM has more high efficiency than other Video-LLMs, which further shows the potential of ICM for quicker training times and less source use,

[4]Due to the space limit, we do not illustrate other metrics on these 11 negative implicit visual sentiment. Actually, ICM is still the best-performing approach.

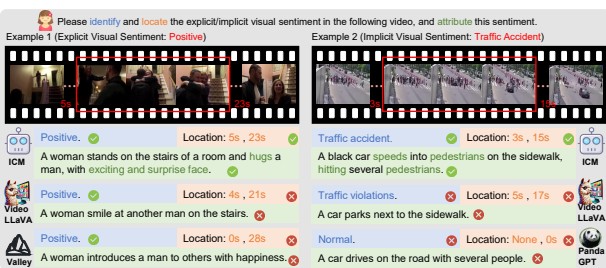

**Figure 5: Two samples to compare ICM with other baselines.**

thereby improving its applicative use in real-world applications. **(2)** ICM shows fast convergence compared to its variants, indicating that the integration of MoE architecture and causal intervention can accelerate the convergence process. **(3)** ICM shows fast convergence compared to without scene-tuning, where the loss is 3.95 at the convergence fitting point. This again justifies the importance of scene understanding before Omni-SILA tuning.

## 5.6 Qualitative Analysis

As illustrated in Figure 5, we provide a qualitative analysis to intuitively compare the performance of ICM with other Video-LLMs on the Omni-SILA task. Specifically, we randomly select two samples from each of explicit and implicit Omni-SILA datasets, asking these approaches to "*Identify and locate the visual sentiment in the following video, and attribute this sentiment*". Due to the space limit, we choose the top-3 well-performing approaches. From this figure, we can see that: **(1)** Identifying and locating implicit visual sentiment is more challenging than the explicit. For instance, Video-LLaVA can roughly locate and precisely identify *positive* sentiment in Example 1, but has difficulties in locating *traffic accident* in Example 2. However, ICM can precisely identify and locate the *traffic accident*. **(2)** Attributing both explicit and implicit visual sentiments is challenging. All the advanced Video-LLMs are difficult to attribute visual sentiments, even some approaches are nonsense. While ICM can provide reasonable attributions due to the capture of omni-scene information, such as *surprise* face, *hugging* action in Example 1, and *speeds, hitting* action, *pedestrians* visual background in Example 2. This again justifies the importance of omni-scene information, and effectiveness of ICM for capturing such information.

## 6 Conclusion

In this paper, we address a new Omni-SILA task, aiming to identify what is the visual sentiment, locate when it occurs and attribute why this sentiment in videos. In particular, we propose an ICM approach to address this task by leveraging both explicit and implicit scene information. The core components of ICM involve a SBM block and an IEC block to effectively model scene information and highlight the implicit scene information beyond the explicit. Experimental results on our constructed explicit and implicit Omni-SILA datasets demonstrate the superior performance of ICM over several advanced Video-LLMs. In our future work, we would like to train a Video-LLM supporting more signals like audio from scratch to further boost the performance of sentiment identifying, locating and attributing in videos. In addition, we would like to leverage some light-weighting technologies (e.g., LLM distillation and compression) to further improve the efficiency of our ICM apporach.

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
