# OpenReview forum: "Omni-SILA: Towards Omni-scene Driven Visual Sentiment Identifying, Locating and Attributing in Videos"
_ACM.org/TheWebConf/2025/Conference — WWW 2025 Oral_

### Official Review · Reviewer_ZJJc · 2024-12-01

**Novelty:** 6
**Technical Quality:** 6

**Review:**

**Pros:**

- The study presents a novel problem formulation that extends visual sentiment analysis to incorporate implicit scene information and attribution.
- It features a well-structured technical approach that combines Mixture of Experts (MoE) architecture with causal interventions.
- The experimental validation is comprehensive, including ablation studies and qualitative analysis.
- The research has strong practical relevance for applications in content moderation and safety.
- There is good attention given to efficiency aspects in addition to effectiveness.

**Cons:**

- There is limited discussion regarding failure cases and the study's limitations.
- Details on dataset construction could be more thorough.
- Information about computational requirements and training processes is somewhat sparse.
- Privacy and ethical considerations surrounding surveillance video analysis are not explored in depth.
- Some technical choices, such as the selection of expert architectures, could be better justified.

**Questions:**

1. How does the system handle cases where explicit and implicit signals conflict significantly, such as a smiling face during a violent action? Is there a principled method for resolving such conflicts?

2. The paper demonstrates good results in analyzing surveillance video. Have you tested the approach in other domains, such as social media videos or movie clips, which may have different characteristics?

3. Regarding the Scene-Balanced Mixture of Experts (MoE), how sensitive is the performance to the number of experts and the routing strategy used? What considerations led to the decision to choose four experts?

**Reviewer Confidence:**

3: The reviewer is confident but not certain that the evaluation is correct

**Scope:**

4: The work is relevant to the Web and to the track, and is of broad interest to the community

---

### Official Review · Reviewer_1oGm · 2024-12-03

**Novelty:** 6
**Technical Quality:** 5

**Review:**

This paper presents a new task, Omni-scene driven visual Sentiment Identifying, Locating and Attributing in videos (Omni-SILA), which aims to comprehensively analyse visual sentiment in videos, going beyond simply identifying the sentiment to also locate where it occurs and attribute its cause. This task is particularly relevant considering the increasing prevalence of video content and the need for a deeper understanding of the emotional nuances conveyed through this medium. The authors construct explicit and implicit Omni-SILA datasets to evaluate the effectiveness of their proposed approach, addressing a gap in existing research.
The authors propose an Implicit-enhanced Causal MoE (ICM) approach to address the Omni-SILA task, tackling the challenges of modelling scene information and highlighting implicit scene information beyond explicit cues like facial expressions. The ICM framework effectively integrates and adapts existing techniques, Scene-Balanced MoE (SBM) and Implicit-Enhanced Causal (IEC).

**Questions:**

1. The paper acknowledges the computational cost of the ICM approach. Have the authors explored any strategies for optimising the efficiency of the model, such as model compression or distillation techniques? This would be beneficial for making the approach more practical for real-world applications.

2. The evaluation primarily focuses on quantitative metrics. Could the authors provide more qualitative examples (instead of selecting two samples) to illustrate the specific strengths of the ICM approach in highlighting different aspects of visual sentiment analysis? This would further enhance the understanding of the model's capabilities and its potential applications.

3. The implicit Omni-SILA dataset focuses on audio-free surveillance video. While this is a valuable contribution, how do the authors envision extending the ICM approach to incorporate audio information? Could the framework be adapted to analyse multimodal sentiment in videos with audio, considering that audio cues can significantly influence the perception of sentiment?

**Reviewer Confidence:**

3: The reviewer is confident but not certain that the evaluation is correct

**Scope:**

3: The work is somewhat relevant to the Web and to the track, and is of narrow interest to a sub-community

---

### Official Review · Reviewer_jDtU · 2024-12-04

**Novelty:** 6
**Technical Quality:** 6

**Review:**

Strength
1. This work proposes a reasonable approach to the visual sentiment identification task.
  1. On a higher level, the system ensembles several scene-understanding modules.
  2. On a lower level, there is scene-balance MoE (SBM) to fuse the aforementioned modules organically, and an Implicit-Enhanced Causal (IEC) block to prevent the model from taking shortcuts.
2. There are extensive experiments to justify the effectiveness of the proposed approach, including:
  1. Analysis of SBM, IEC, and scene-tuning,
  2. Ablation study over the four scene modules.
3. They created an Omni-SILA dataset, which is another contribution to the specific field.

Weakness
1. In Figure 2, it is not clear what the FEE, HAE, etc. are. I suggest renaming "XX Modeling" to "XX Expert" in the SEM block.
2. It is not clear if repeated experiments is conducted, and what is the variance of the experiments

**Questions:**

1. Will the  Omni-SILA  be made public?
2. What is the distribution of the scene router after training?

**Reviewer Confidence:**

3: The reviewer is confident but not certain that the evaluation is correct

**Scope:**

3: The work is somewhat relevant to the Web and to the track, and is of narrow interest to a sub-community

---

### Official Review · Reviewer_47ta · 2024-12-04

**Novelty:** 5
**Technical Quality:** 5

**Review:**

**Summary**

The paper introduces a new task Omni-SILA, which focuses on identifying, localizing, and attributing visual sentiments in videos by leveraging both explicit and implicit scene information. This paper proposes an ICM approach for addressing the Omni-SILA task, where SBM models explicit and implicit scene information, and IEC mitigates bias and highlights implicit scene details using causal intervention. The framework is tested on a newly constructed Omni-SILA dataset with experiments showing significant improvements over existing baselines.

**Strengths**

1. The paper is well-written and easy to understand.
2. The paper addresses a gap in visual sentiment understanding by combining identification, localization, and attribution with implicit and explicit scene integration.
3. Well-designed experiments demonstrate the importance of SBM and IEC, providing a clear understanding of their contributions.

**Weaknesses**
1. The Causal Intervention Graph session needs a better explanation. The method to differentiate implicit from explicit scenes using causal intervention isn't well-elaborated. It's unclear how the implicit bias is quantitatively measured or addressed.
2. How to find the invisible confounding factors C? I didn’t see a clear link showing that reducing the bias in modeling between X and C can hide the explicit scene information.
3. Instruction tuning can lead to catastrophic forgetting. The paper pre-tuned on 4 datasets. It’s true that it can improve downstream task performance, but it will also decrease the model’s generalizability.
4. The paper mentions that "the output dimension d is different, which will force the model to pay much attention to scene expert with high dimensions" in line 374. A dimension reduction method can help. Just averaging might cause some information to be lost.

**Questions:**

1. Eq (5) and Eq (6) are not consistent with line 423. How did you get these equations from that line? Since $y_x$  doesn’t include $m$.
2. It’s unclear how to find m, p, q in line 425.
3. How to calculate $P_j$ from P?

**Reviewer Confidence:**

3: The reviewer is confident but not certain that the evaluation is correct

**Scope:**

3: The work is somewhat relevant to the Web and to the track, and is of narrow interest to a sub-community